# Altered Transcription Factor Binding and Gene Bivalency in Islets of Intrauterine Growth Retarded Rats

**DOI:** 10.3390/cells9061435

**Published:** 2020-06-09

**Authors:** Yu-Chin Lien, Paul Zhiping Wang, Xueqing Maggie Lu, Rebecca A. Simmons

**Affiliations:** 1Center for Research on Reproduction and Women’s Health, Perelman School of Medicine, The University of Pennsylvania, Philadelphia, PA 19104, USA; ylien@pennmedicine.upenn.edu; 2Division of Neonatology, Department of Pediatrics, Children’s Hospital of Philadelphia, Philadelphia, PA 19104, USA; 3Institute for Biomedical Informatics, Perelman School of Medicine, The University of Pennsylvania, Philadelphia, PA 19104, USA; wangpaul@pennmedicine.upenn.edu (P.Z.W.); xueqing.lu@pennmedicine.upenn.edu (X.M.L.)

**Keywords:** islets, histone modifications, ChIP-Seq, intrauterine growth restriction, epigenetics, transcription factor binding motif, bivalent gene

## Abstract

Intrauterine growth retardation (IUGR), which induces epigenetic modifications and permanent changes in gene expression, has been associated with the development of type 2 diabetes. Using a rat model of IUGR, we performed ChIP-Seq to identify and map genome-wide histone modifications and gene dysregulation in islets from 2- and 10-week rats. IUGR induced significant changes in the enrichment of H3K4me3, H3K27me3, and H3K27Ac marks in both 2-wk and 10-wk islets, which were correlated with expression changes of multiple genes critical for islet function in IUGR islets. ChIP-Seq analysis showed that IUGR-induced histone mark changes were enriched at critical transcription factor binding motifs, such as C/EBPs, Ets1, Bcl6, Thrb, Ebf1, Sox9, and Mitf. These transcription factors were also identified as top upstream regulators in our previously published transcriptome study. In addition, our ChIP-seq data revealed more than 1000 potential bivalent genes as identified by enrichment of both H3K4me3 and H3K27me3. The poised state of many potential bivalent genes was altered by IUGR, particularly *Acod1, Fgf21, Serpina11, Cdh16, Lrrc27,* and *Lrrc66,* key islet genes. Collectively, our findings suggest alterations of histone modification in key transcription factors and genes that may contribute to long-term gene dysregulation and an abnormal islet phenotype in IUGR rats.

## 1. Introduction

Poor fetal growth with low birth weight due to intrauterine growth restriction (IUGR) is strongly associated with increased risk of adulthood type 2 diabetes (T2D) in human and animal studies [1,2,3,4]. To elucidate the mechanisms by which IUGR results in the development of diabetes, we have developed a rat model of uteroplacental insufficiency induced by bilateral uterine artery ligation, which reduces the blood supply and critical substrates to the fetus [3,5]. This adverse intrauterine milieu impairs the development of pancreatic islets in the fetus and causes permanent *β*-cell dysfunction [3,5]. Immune cell infiltration in the pancreas and decreased islet capillary density are observed prior to birth in IUGR rats [6]. In the newborn period, the IUGR pups exhibit impaired insulin secretion and reduced islet vascularity. By 10 weeks of age, *β*-cell mass is reduced and IUGR rats have developed mild-fasting hyperglycemia and exhibit diminished glucose- and leucine-stimulated insulin secretion [3,6,7,8]. To investigate the global transcriptome changes and elucidate the mechanisms that may mediate *β*-cell dysfunction in IUGR rats, we performed a RNA-Seq study in islets isolated from 2-week and 10-week rats [9]. Thousands of genes in pathways regulating islet function were altered in IUGR rats, including pathways modulating nutrient metabolism and transport, insulin secretion, mitochondrial function and oxidative stress, extracellular matrix, angiogenesis, innervation, immune function, and inflammation. Changes in these pathways strongly correlated with their relevant phenotypic changes.

Epigenetic modifications of the genome are heritable changes affecting gene expression and allow the stable propagation of gene activity states from one generation of cells to the next [10,11,12]. Epigenetic states can be modified by environmental factors, which may contribute to the development of abnormal phenotypes and diseases. Epigenetic mechanisms have been implicated to play a critical role in many metabolic disorders, such as diabetes, obesity, and cardiovascular diseases [13,14,15,16,17]. Using our IUGR model, we demonstrated that the expression of *Pdx1*, a pancreatic and duodenal homeobox 1 transcription factor critical for β-function and development, is epigenetically silenced in IUGR rats [18]. A short treatment course of exendin-4, a long-acting glucagon-like peptide-1 (GLP-1) analogue, in the newborn period reverses epigenetic modifications at the proximal promoter of *Pdx1* and prevents the development of diabetes in IUGR rats [7,19]. These studies strongly implicate epigenetic mechanisms leading to the development of *β*-cell dysfunction in IUGR animals.

There are at least three distinct categories of epigenetic modifications: histone modifications, DNA methylation, and non-coding RNAs [20,21,22]. We have previously demonstrated that IUGR alters DNA methylation at ~1400 loci in adult rat islets, and many of these changes are associated with nearby gene expression changes and long term susceptibility to T2D [23]. However, whether histone modifications also play a role in IUGR-induced gene dysregulation in islets is still unclear. The amino termini of histones can be modified by acetylation, methylation, phosphorylation, ubiquitylation, sumoylation, glycosylation, and ADP ribosylation [10,21]. The most common histone modifications are acetylation and methylation of lysine (K) residues of histone 3 (H3) and histone 4 (H4). Increased acetylation is associated with transcription activation, such as acetylation of H3K27, whereas decreased acetylation usually induces transcription repression. In contrast, methylation of histones can be associated with both transcription repression and activation. Trimethylation of H3K4 is implicated in the activation of transcription, whereas trimethylation of H3K27 results in transcriptional silencing.

In the current study, we present the first genome-wide assessment of histone modifications in both 2-week and 10-week rat islets and assess IUGR-induced changes of H3K4 trimethylation (H3K4me3), H3K27 trimethylation (H3K27me3), and H3K27 acetylation (H3K27Ac). We have demonstrated that alterations of histone modifications in key transcription factor binding motifs and genes are associated with changes in expression of key genes, suggesting that these epigenetic modifications contribute to long-term gene dysregulation and an abnormal islet phenotype in IUGR rats.

## 2. Materials and Methods

### 2.1. Animal Model

The animals and procedures used in this study were approved by the Animal Care Committee of the Children’s Hospital of Philadelphia and the Perelman School of Medicine at the University of Pennsylvania. Our IUGR animal model has been previously described [3]. Briefly, bilateral uterine artery ligation was performed in pregnant Sprague Dawley rats (Charles River, Raleigh, NC, USA) at E18.5. Sham surgery was performed in controls. Pups were weighed on the day of delivery to confirm intrauterine growth restriction, and litters were randomly culled to 8 to equilibrate postnatal nutrient availability. Dams and offspring were given ad libitum access to water and standard rodent chow.

### 2.2. Islet Isolation

Pancreata were excised at 2 and 10 weeks of age, and followed by islet isolation. Following the previous phenotypic studies using pooled samples from both sexes at 2 weeks of age [6,9], 4 pancreata per litter from both sexes were pooled for each 2-week sample. 2 pancreata per litter from male rats were pooled for each 10-week sample since the diabetic phenotype in adulthood is male sex-specific [6]. Pancreatic islets were isolated as previously described [6]. Briefly, pancreata were digested with Collagenase P (Millipore Sigma, St. Louis, MO, USA) in HBSS supplemented with 4 mM NaCO_3_ and 1% BSA for 15 min at 37 °C. Digested tissues were then washed in cold supplemented HBSS solution without collagenase. Islets were isolated by histopaque gradient centrifugation.

### 2.3. Chromatin Preparation and Chromatin Immunoprecipitation (ChIP)

Chromatin was extracted from freshly isolated islets and ChIP with antibodies for H3K4me3 (07-473, Millipore Sigma, St. Louis, MO, USA), H3K27me3 (07-449, Millipore Sigma, St. Louis, MO, USA), and H3K27Ac (ab4729, Abcam, Cambridge, MA, USA) and prepared as previously described [24]. Briefly, islets were cross-linked with 2.22% formaldehyde in PBS for 10 min at room temperature followed by incubation with 0.14 M glycine to stop cross-linking. Samples were sonicated using a BioRuptor (Diagenode, Denville, NJ, USA) to shear the chromatin with a high setting of cycling 30 s on −30 s off for 5 min. The number of cycles to achieve appropriate shearing was monitored using a 2100 BioAnalyzer with a high sensitivity DNA kit (Agilent, Santa Clara, CA, USA). 4–5 µg of chromatin was used for ChIP with antibodies for histone modifications. DNA from un-crosslinked chromatin was purified using Qiagen PCR purification kit (Qiagen, Germantown, MD, USA).

### 2.4. ChIP-Seq and Data Analysis

Multiplexed ChIP-Seq libraries were created using the Illumina ChIP-seq DNA sample prep kit (Illumina, San Diego, CA, USA) following the manufacturer’s instructions and a previously published protocol [24]. Three biological replicates for each group were used to prepare libraries. Libraries for H3K4me3, H3K27me3, H3K27Ac, and input were sequenced to 50 bp on an Illumina hiSeq2000. FASTQC was used to check the raw sequencing data quality, and adapters were trimmed from the sequence reads by Trimmomatic [25]. The reads were mapped to rat genome assembly rn6 using STAR [26] with default settings, then followed by filtering with samtools [27] to remove unmapped reads, reads mapped to multiple locations, and mitochondria reads. Histone enriched sites were called by MACS2 [28] using a false discovery rate (FDR, q-value) cut-off of 0.05, and histone marker differential enrichment analysis was performed with diffReps [29]. All sequencing tracks were made by HOMER [30]. Functional analysis was performed using QIAGEN’s Ingenuity^®^ pathway analysis (IPA^®^). Enrichment analysis of transcription factor binding motifs was carried out using HOMER. Bivalent domains were identified by the enrichment of both H3K4me3 and H3K27me3 marks in which enrichment peaks have at least 5 bp overlap [31]. Genome browser tracks were generated by first merging all replicates from the same condition, then normalized by sequence depth and input value. The Log2 ratio of sample to inputs is reported and visualized on Integrative Genomics Viewer (IGV) genome browser (http://software.broadinstitute.org/software/igv/). Sequence data have been deposited in NCBI’s Gene Expression Omnibus and are accessible through GEO Series accession number GSE147992.

## 3. Results

### 3.1. ChIP-Seq Analysis

We performed ChIP-Seq analysis for H3K4me3, H3K27me3, and H3K27Ac histone modifications on pancreatic islets isolated from 2 and 10 weeks old control and IUGR Sprague Dawley rats. All ChIP-Seq data aligned well with the reference genome (rn6 assembly). The alignment rates ranged from 70.8% to 87.5%. Principal component analysis (PCA plot, Figure 1) showed a clear separation for each histone mark, indicating the unique result for different modifications. We annotated these loci enriched with histone marks by proximity to RefSeq and known genes. On average, about 82% of H3K27me3 enriched loci and 78% of H3K27Ac enriched loci were >5 kb from the nearest gene (Appendix A). This is consistent with previous findings showing that both H3K27me3 and H3K27Ac were located in distal and proximal regions from transcription start sites (TSS), and also mark enhancer and regulatory regions [32,33,34]. Surprisingly, 75% of H3K4me3 enriched loci were located >5 kb from the nearest gene (Appendix A), indicating a large number of potentially active novel transcriptional start sites in islets. The genomic locations of these histone mark enriched loci are shown in Figure 2. Interestingly, less than 10% of modifications were located in promoters and transcriptional start sites (TSS). The majority of the marks were located in intergenic or intronic regions. This was especially true for H3K27me3, and approximately 70% of enriched loci were located in intergenic regions. These may also represent many potential novel TSS, TSS for non-coding RNAs, and regulatory regions.

Integrating ChIP-Seq results with our previously published RNA-Seq data [9], we found no direct association between levels of normalized gene expression and genomic location of histone marks (Appendix A). However, as expected, enrichment of H3K27me3 marks were associated with lower levels of gene expression regardless of their genomic location. To assess the distribution and enrichment of each histone mark in control vs. IUGR islets, we created transcription start site plots (TSS plots, Figure 3). The TSS plots showed higher enrichment of H3K4me3 and H3K27Ac in control compared to IUGR islets at both 2-wk and 10-wk, which was consistent with the RNA-Seq data showing far more genes that were down-regulated in IUGR islets [9]. Enrichment of H3K27me3 was higher in 10 wk IUGR islets near promoters/transcription start sites, but higher in control islets beyond these regions, indicating potential regulatory regions.

### 3.2. IUGR-Induced Histone Modification Changes in Islets Correlated with Transcription Activities and Phenotypes

To avoid the complexity of gene overlapping, we focused on the alterations of histone marks within 5 kb of transcriptional start sites. Thousands of genes showed histone mark changes in both 2-wk and 10-wk IUGR islets (Appendix A). Integrating our ChIP-Seq and RNA-Seq datasets, we found histone mark changes at many genes in IUGR islets that correlated with their RNA expression (Table 1, Appendix A), suggesting these genes were regulated by histone modifications. Interestingly, we found 6 genes potentially regulated by all 3 histone modifications in 10-wk islets, in which enrichment of 3 histone marks correlated with expected changes in gene expression in IUGR islets (Table 2, Appendix A). These included 3 up-regulated genes, *Trpm5, Tfam, and Mcf2l,* and 3 down-regulated genes, *Slc28a2, Tnf*, and *Mpz*. Transient receptor potential cation channel, subfamily M, member 5 (*Trpm5*) is a calcium-activated non-selective cation channel. It is expressed in β-cells and regulates glucose-stimulated insulin secretion [35,36]. Mitochondrial transcription factor A (*Tfam*) is a key regulator for mitochondrial genome replication and gene transcription [37]. It is a target gene of Pdx1 and plays a critical role in maintaining normal mitochondrial DNA (mtDNA) copy number, ATP production, and glucose-stimulated insulin secretion (GSIS) [38,39]. This is a particularly relevant finding, as the mitochondrial function is markedly impaired in IUGR islets and mtDNA levels are decreased [40]. MCF.2 cell line derived transforming sequence like (*Mcf2l*) encodes a guanine nucleotide exchange factor DBS that can interact specifically with the GTP-bound Rac1 and plays a role in the Rho/Rac signaling pathways [41]. These small G-proteins regulate cytoskeletal remodeling, vesicular transport, and GSIS, and play critical roles in normal β-cell function [41,42,43]. Solute carrier family 28 member 2 (*Slc28a2*) encodes a purine-specific concentrative nucleoside transporter, CNT2 [44], and is important for controlling extracellular adenosine levels [45]. Adenosine signaling plays a critical role in the regulation of glucose homeostasis, insulin sensitivity, and the pathogenesis of type 1 and type 2 diabetes [45,46]. Tumor necrosis factor (*Tnf*) is an important cytokine regulating immune function and inflammation. Myelin protein zero (*Mpz*) is a major structural protein of the peripheral myelin sheath and is necessary for normal neuronal function [47]. Pancreatic islets are innervated with sympathetic, parasympathetic, and sensory nervous systems, and they play important roles in regulating islet hormone secretion and glucose metabolism [48,49]. The gene expression and histone modifications changes of these 6 genes correlate well with phenotypes that we have observed in IUGR islets [3,7].

We also identified multiple differentially expressed genes with decreased H3K27Ac modifications in 2-wk IUGR islets, and the changes persisted until 10 weeks of age. These genes included *Dram1, Foxa3, Gpt2, Ppard, Pycr1, Sel1l, Tst*, and *Ulk1* (Table 3). The decreased expression of these genes was consistent with decreased H3K27Ac enrichment. DNA-damage regulated autophagy modulator 1 (*Dram1*) and unc-51 like autophagy activating kinase 1 (*Ulk1*) are two genes regulating autophagy. Autophagy is important for normal islet function [50]. Both *Dram1* and *Ulk1* regulate autophagy via modulating mTOR signaling [51,52]. Glutamic-pyruvic transaminase 2 (*Gpt2*), Pyrroline-5-carboxylate reductase 1 (*Pycr1*), and thiosulfate sulfurtransferase (*Tst*) are three enzymes important for mitochondrial function. *Gpt2* catalyzes the reversible transamination between alanine and 2-oxoglutarate to generate pyruvate and glutamate, and is pivotal to metabolic adaptation [53]. *Pycr1* catalyzes the conversion of pyrroline-5-carboxylate to proline, and is important for amino acid metabolism, oxidative stress regulation, and mitochondrial integrity [54]. *Tst* is essential for the import of 5S ribosomal RNA to mitochondria [55]. It is also a beneficial regulator of insulin sensitivity and mitochondrial function [56]. *Sel1l*, a component of the endoplasmic reticulum-associated degradation (ERAD) pathway, regulates glucose-stimulated insulin secretion and insulin trafficking via modulating integrin and extracellular matrix [57]. Forkhead box protein A3 (*Foxa3*, a.k.a. hepatocyte nuclear factor 3-γ) regulates the expression of glucose transporter 2 (*Glut2)*, plays a role in the maintenance of glucose homeostasis, and also modulates innate immunity [58,59]. Peroxisome proliferator-activated receptor delta (*Ppard*) is critical in regulating insulin sensitivity, glucose metabolism, and fatty acid β-oxidation [60].

To identify molecular pathways potentially regulated by histone modifications that were disrupted in 10-wk IUGR islets, Ingenuity Pathway Analysis (IPA) was used to map histone mark alterations at identified gene loci into functional networks. IPA analysis of genes predicted to be regulated by histone modifications revealed enrichment for pathways critical for islet function (Table 4), such as type 2 diabetes signaling, tight junction signaling, Wnt/*β*-catenin signaling, and mitochondrial function. There was also enrichment for pathways that regulate T-cell, B-cell, and macrophage function, and pathways modulating innervation and neuronal function, including axonal guidance signaling and c-AMP signaling. The pathways regulating angiogenesis and vascular remodeling were also altered in 10-wk IUGR islets, including PDGF signaling and phospholipase C signaling. Of note, these findings overlapped with key pathways identified by our RNA-Seq data and were also highly correlated with IUGR phenotypes [9].

### 3.3. IUGR Altered Histone Modifications at Critical Transcription Factor Binding Motifs

#### 3.3.1. Two-Week-Old Animals

To further investigate whether changes in histone modification in IUGR islets were located near transcription factor binding sites that may facilitate or block transcription factor binding and result in gene dysregulation in IUGR, we performed HOMER binding motif analysis for all 3 histone marks. The top binding motifs for each histone modification are listed in Appendix A. The top binding motifs associated with H3K4me3 enrichment in 2-wk control islets included many transcription factors important for β-cell development and function, such as homeobox protein Nkx6.1, Pdx1, Forkhead box protein M1 (Foxm1), and Foxa3, and transcription factors regulating cell pluripotency, such as homeobox protein Nanog and octamer-binding transcription factor-4 (Oct4) (Appendix A) [61,62]. However, in 2-wk IUGR islets, the top binding motifs for H3K4me3 were transcription factors belonging to interferon regulatory factor (IRF) family and PR domain zinc finger protein 1 (Prdm1), a transcription factor regulating interferon-β expression and B-cell maturation [63,64], suggesting up-regulation of genes modulating immune responses in 2-wk IUGR islets. This is consistent with our previous data showing marked inflammation in IUGR islets [6].

Additional binding motifs that were enriched with H3K4me3 in IUGR islets include FoxL2, ETS like 1 (Elk1), ETS proto-oncogene 1 (Ets1), ETS variant 1 (Etv1), neurofibromin 1 (Nf1), and hepatocyte nuclear factor 1homeobox B (Hnf1b) (Figure 4a), which was consistent with upregulation of their target genes which we previously demonstrated [9]. Elk1, Ets1, Etv1, and Fli1 are transcription factors belonging to E26 transformation-specific (ETS) family [65]. They regulate many processes during development, such as cell proliferation, differentiation, migration, apoptosis, and mesenchymal-epithelial interactions [66]. The ETS family also plays a critical role during pancreatic development [67]. Increased ETS1 expression is associated with β-cell dysfunction induced by glucotoxicity in type 2 diabetes [68]. NF1 is a negative regulator of the Ras pathway and controls cellular growth and neural development [69]. HNF1b is crucial in modulating pancreatic multipotent progenitor cells and generation of endocrine precursors and is associated with maturity-onset of Diabetes of the Young-5 (MODY5) [70]. Tight control of its expression is important to maintain insulin secretion and β-cell viability [71].

The top binding motifs associated with H3K27Ac enrichment for both 2-wk control and IUGR islets included androgen receptor (AR), Foxo1, thyroid hormone receptor-β (Thrb), ETS-related gene (Erg), and ETS variant 2 (Etv2) (Appendix A). Friend leukemia integration 1 transcription factor (Fli1) and paired-like homeodomain 1 (Pitx1) were among the top binding motifs in 2-wk IUGR islets.

Finally, the top three binding motifs associated with H3K27me3 in 2-wk IUGR islets were insulin gene enhancer protein ISL-1, Nkx6.1, and zinc finger protein ZBTB7A (Appendix A). Other top binding motifs with altered H3K27me3 enrichment in 2-wk IUGR islets were Nkx2.5, Nkx2.1, B-cell lymphoma 6 protein (Bcl6), paired box 7 (Pax7), Sox4, and CCAAT enhancer-binding proteins (C/EBPs) (Figure 4b). While Nkx2.5, Nkx2.1, and Bcl6 binding motifs were enriched with H3K27me3 in IUGR islets, Pax7, Sox4, and C/EBP binding motifs lost their H3K27me3 marks. Bcl6 plays a role in lipid metabolism [72]. It also modulates cell inflammatory responses [73]. C/EBPs play important roles in normal β-cell function, while C/EBP-δ is anti-apoptotic and anti-inflammatory in β-cells [74], overexpression of C/EBP-β is associated with endoplasmic reticulum (ER) stress and β-cell failure [75]. C/EBP-β also inhibits the expression of the insulin gene in β-cells [76]. Changes in enrichment at these key transcription factors is associated with changes in expression of their target genes, indicating a key regulatory role of histone modifications in regulating gene expression in islets.

#### 3.3.2. Ten-Week-Old Animals

At 10 weeks of age, the top binding motifs enriched with H3K4me3 in both control and IUGR islets were IRFs, Foxa1, LIM/homeobox protein 3 (Lhx3), and Nkx6.1 (Appendix A). Other binding motifs with altered H3K4me3 enrichment in 10-wk IUGR islets included Fli1, Ets1, caudal type homeobox 2 (Cdx2), Sox10, and Sox3 (Figure 5a). Cdx2, Sox10, and Sox3 binding motifs in 10-wk IUGR islets were enriched with H3K4me3 modifications. In contrast, Fli1 and Ets1 binding motifs lost their H3K4me3 marks in IUGR islets.

Neurogenic differentiation 1 (NeuroD1), neurogenin-2 (NeuroG2), Foxa1, Foxo1, protein atonal homolog 1 (Atoh1), and oligodendrocyte transcription factor 2 (Olig2) were among the top binding motifs associated with H3K27Ac (Appendix A), and there were no clear differences between 10-wk control and IUGR islets at these loci.

In 10-wk IUGR islets, several transcription factors in the SRY-related HMG-box (Sox) family were among the top binding motifs enriched with H3K27me3, including Sox3, 6, 10, and 17 (Appendix A), which play important roles in regulating cell fate, neuronal development, angiogenesis, and Wnt signaling [77,78,79]. We observed decreased expression of their target genes in IUGR rats that may be responsible for the decreased innervation and vascularity, which we have observed in previous studies [6,8]. The top binding motifs with altered H3K27me3 marks in 10-wk IUGR islets were Myb proto-oncogene (Myb), Thrb, early B-cell factor 1 (Ebf1), Sox9, microphthalmia-associated transcription factor (Mitf), Oct4-Sox2-TCD-Nanog, and pituitary-specific transcription factor 1 (Pit1, also known as Pou1f1) (Figure 5b). Ebf1, Sox9, Mitf, Oct4-Sox2-TCD-Nanog, and Pit1 binding motifs in 10-wk IUGR islets were enriched with H3K27me3 modifications, however, Myb and Thrb binding motifs lost their H3K27me3 marks in IUGR islets. Thrb mediates the biological activities of the thyroid hormone, which plays a critical role in energy metabolism, as well as islet cell development, differentiation, maturation, and function [80]. Ebf1 is a key transcription factor for B-cell differentiation and function [81]. It also controls neuronal differentiation and migration during neurogenesis [82]. Mitf can modulate β-cell function by controlling insulin secretion and paired box 6 (Pax6) transcription [83].

Interestingly, Lhx1, 2, and 3 were among the top binding motifs associated with H3K27me3 in both control and IUGR islets at both ages. Lhx1, 2, and 3 are LIM-HD transcription factors related to Isl1, and are critical for regulating organogenesis and development of neuronal and hematopoietic systems [84], which are also important for normal islet function. Lhx1 has been shown to act as a novel regulator of islet function, which interacting with Isl1 and LIM domain-binding protein 1 (Ldb1) and maintaining glucose homeostasis via regulating the expression of glucagon-like peptide 1 receptor (*Glp1R*) [85]. Indeed, Lhx1 was predicted as an inhibited upstream regulator in both 2-wk and 10-wk IUGR islets in our transcriptome datasets [9]. In addition to H3K27me3, Lhx1 binding motifs were also associated with H3K4me3 enrichment (Appendix A). Thus, the fine balance between repression and activation marks at these loci may regulate the expression of Lhx1 target genes and contribute to the phenotype in IUGR islets.

#### 3.3.3. Persistent Changes from Two- to Ten-Week-Old Animals

We identified several binding motifs with altered H3K4me3 marks in 2-wk IUGR islets, which persisted until 10 weeks of age, including ELF3, ELF5, ERG, ETS1, ETV1, ETV2, EWS:ERG, GABPA, and SMAD4 (Table 5). All of these binding motifs exhibited increased H3K4me3 modifications in IUGR islets. E74 like ETS transcription factor 3 (ELF3) and E74 like ETS transcription factor 5 (ELF5) are two transcription factors belonging to the ETS family. They also play a critical role during pancreatic development similar to other ETS family transcription factors, particularly in regulating epithelium-specific gene expression [67]. GA binding protein transcription factor subunit alpha (GABPA) is also an ETS family transcription factor. GABP regulates the expression of cytochrome oxidase and mitochondrial import protein TIMM23, and is required for mitochondrial biogenesis and normal function [86,87]. SMAD family member 4 (SMAD4) is a transcription factor mediating transforming growth factor- β (TGF-β) signaling, which is critical for pancreas development and function [88,89].

MEIS1, PR, and Olig2 were the binding motifs with altered H3K27me3 enrichment in both 2-wk and 10-wk IUGR islets (Table 5). All of them exhibited decreased H3K27me3 modifications in IUGR islets. Meis homeobox 1 (MEIS1) is essential for embryonic development of hematopoietic and vascular system [90]. It also regulates the expression of paired box 6 (Pax6), which plays an important role for islet development and hormone production [91]. Progesterone receptor (PR) mediates progesterone signaling, which regulates glucose, lipid, and protein metabolism [92]. The blockage of PR signaling may enhance β-cell viability and increase β-cell proliferation and insulin secretion [93,94]. Olig2 is critical for the differentiation of oligodendrocytes, which are important for myelination of axons [95,96]. Olig2 also plays a role in neural repair [97].

Consistent with the HOMER binding motif analysis, many transcription factors discussed above were also identified as critical upstream regulators modulating gene expression changes observed in IUGR islets in our transcriptome datasets (Table 6).

### 3.4. Poised States of Potential Bivalent Genes Were Altered in IUGR Islets

Bivalent genes, identified by enrichment of both H3K4me3 and H3K27me3 marks in promoter regions, play critical roles in regulating pluripotency of embryonic stem cells, maintaining gene imprinting, and fine-tuning gene expression during development [98,99,100,101]. However, bivalent chromatin also is associated with diseases such as cancers and Huntington’s disease [102,103]. In β-cells, loss of H3K27me3 marks at poised/bivalent domains may contribute to β-cell de-differentiation, dysfunction, and diabetes [104]. Our ChIP-seq data revealed more than 1000 potential bivalent genes. Lack of expression or very low expression of these genes is consistent with a poised state of transcription. The poised state of many of these potential bivalent genes was altered in 2- and 10-wk IUGR islets (Appendix A). Some genes lost their bivalency in IUGR islets, and some gained their bivalency (Appendix A). Gastrin releasing peptide receptor (*Grpr*) and proprotein convertase subtilisin/kexin type 9 (*Pcsk9*) were the only two genes that gained bivalency in 2-wk IUGR islets. Grpr gained the bivalency in 2-wk IUGR islets and maintained a poised state at 10-wks. Grpr is important for normal islet function and mediates the effect of gastrin-releasing peptide (Grp), an islet neuropeptide, and neutrally stimulated insulin and glucagon secretion [105]. Mice with deficient Grpr have impaired glucose tolerance and reduced glucagon-like peptide 1 (GLP-1) and early insulin responses to gastric glucose [106]. Pcsk9 is an escort protein and facilitates other proteins (mainly low density lipoprotein receptor, LDLR) to endosomes/lysosomes for degradation [107,108]. Deficiency of Pcsk9 results in decreased insulin secretion and increased glucose intolerance [109]. Myosin 1a (*Myo1a*) lost its bivalency in 2-wk IUGR islets, and this change was sustained until 10-wks of age. It is an unconventional myosin and functions as actin-based molecular motors in immune cells [110].

Several genes known to be critical in pathways modulating islet function gained de novo bivalency in 10-wk IUGR islets, such as *Ang2, Grm7,* and *Nr0b2.* Angiogenin, ribonuclease A family, member 2 (Ang2) is a potent mediator for new blood vessel formation [111]. Metabotropic glutamate receptor 7 (Grm7) is activated by the excitatory neurotransmitter glutamate, which inhibits the c-AMP cascade and regulates NMDA receptor activity [112]. Glutamate signaling plays a critical role in modulating pancreatic hormone secretion and islet cell function and viability [113]. Small heterodimer partner (Nr0b2) interacts with nuclear receptors, such as estrogen, retinoid, and thyroid hormone receptors inhibit their transcriptional activities and regulate genes involved in multiple metabolic pathways [114]. Overexpression of Nr0b2 can normalize impaired GSIS and enhance glucose sensitivity in uncoupling protein 2 (UCP2)-overexpressed β-cells [115].

Interestingly, we identified six potential bivalent genes critical for islet maturation and long-term IUGR phenotype (Table 7,Appendix A). Three genes, *Acod1, Fgf21*, and *Serpina11*, which were poised in 2-wk control islets, lost their poised state in 10-wk control islets but gained bivalency in 10-wk IUGR islets which was associated with a marked loss of expression of all three genes. In contrast, *Cdh16, Lrrc27*, and *Lrrc66*, which were not poised in control 2-wk islets, gained de novo bivalency in 10-wk control islets, but lost their poised state in 10-wk IUGR islets and expression was increased. Aconitate decarboxylase 1 (Acod1) is an enzyme that catalyzes the decarboxylation of *cis*-aconitate to produce itaconate, and is involved in the inhibition of inflammatory responses [116]. It acts as a negative regulator of the Toll-like receptor (TLR) mediated inflammatory innate response and plays a role in antimicrobial response of innate immune cells. Acod1 and itaconate are also associated with mitochondria and regulate succinate dehydrogenase activity and TCA cycle in macrophages [117,118]. Fibroblast growth factor 21 (Fgf21) is a secreted endocrine factor that functions as a major metabolic regulator [119]. It stimulates glucose uptake in adipocytes via the induction of glucose transporter GLUT1 expression [120]. Fgf21 protects against inflammation and islet hyperplasia induced by a high-fat diet [121]. It also protects against type 2 diabetes in mice by increasing insulin expression and secretion [122]. Loss of *Fgf21* induces insulin resistance and islet dysfunction in mice [123]. Serpin family A member 11 (*Serpina11*) is an uncharacterized antitrypsin-like serine proteinase inhibitor [124]. Cadherin 16 (*Cdh16*) is an atypical cadherin. It functions as the principal mediator for homotypic cell-cell adhesion and plays a role in morphogenesis during tissue development [125]. Leucine-rich repeat-containing 27 (Lrrc27) and leucine-rich repeat-containing 66 (Lrrc66) are proteins belonging to the leucine-rich repeat family of proteins, which play important roles in the development of innate immunity and nervous system [126,127]. They are involved in diverse biological processes, including cell adhesion, cellular trafficking, and hormone-receptor interactions.

## 4. Discussion

Many studies have demonstrated that intrauterine growth restriction is associated with epigenetic changes in different tissues of the offspring [18,128,129,130,131,132,133,134,135,136], and several implicate epigenetic mechanisms in pancreatic β-cell failure and development of T2D [13,15,17,135,137,138,139,140]. In our IUGR model, we have previously demonstrated in adult rats that IUGR induces genome-wide changes in DNA methylation at key loci in islets [23]. However, whether histone modifications are also associated with the persistent gene expression changes observed in IUGR islets had not been fully elucidated yet. To that end, here we present the first ChIP-seq study to assess genome-wide alterations of H3K4me3, H3K27me3, and H3K27Ac marks at both 2-wk and 10-wk IUGR islets since gene expression patterns differ temporally at these two different ages [9]. Consistent with our previous transcriptomic analysis, the changes in histone mark enrichment in IUGR islets mapped to pathways that are important for islet function.

Histone modifications are an important epigenetic mechanism regulating gene expression. Different modifications, as well as which amino acid residue is modified, have different influences on gene transcriptional activity. In the present study, approximately 80% of H3K27me3 and H3K27Ac marks were enriched in the loci >5 kb from the nearest genes. Less than 10% of differentially enriched loci were located in promoters and TSS. This is consistent with previous findings showing that both H3K27me3 and H3K27Ac also mark enhancer/regulatory regions [32,33,34]. While H3K27Ac marks active enhancers, H3K27me3 marks poised enhancers. Much to our surprise, although there were more genes down-regulated rather than up-regulated in IUGR islets, enrichment of H3K27me3 was higher in 2-wk control islets compared with IUGR islets. Beyond promoter/transcription start sites, enrichment of H3K27me3 was also higher in 10-wk control islets. H3K27me3 is generally associated with the repression of transcription, however, Young et al. have identified three enrichment profiles of H3K27me3 with distinct regulatory consequences: a broad domain of H3K27me3 enrichment corresponding to inhibitory of transcription, a peak enrichment around TSS associated with bivalent genes, and a peak enrichment in promoters associated with active transcription [141]. This may partially explain our observation that the enrichment of H3K27me3 was higher in control islets.

Our ChIP-seq analysis of the IUGR islets identified marked changes at pertinent genes and signaling pathways that could underlie β-cell and islet abnormalities associated with uteroplacental insufficiency. Histone modifications at genes that regulate tight junction, Wnt/β-catenin signaling, mitochondrial function, immune responses, neuronal function/innervation, and angiogenesis were significantly altered in 10-wk IUGR islets, which was consistent with observed IUGR phenotypes and reinforced our findings in our previous transcriptome studies [3,6,7,9,40]. Intact islet cell architecture and β-cell to β-cell interaction by intercellular junctions are important for glucose-stimulated insulin secretion [142,143]. Tight junctions may help to segregate the membrane portions enriched with hormone receptors and glucose transporters from those enriched with insulin-containing secretory granules [144,145]. Wnt/β-catenin signaling is important for pancreatic development and mature β-cell function. It also plays a critical role in glucose metabolism and energy homeostasis [146]. β-cell-specific inactivation of β-catenin results in reduced β-cell mass and proliferation, as well as glucose intolerance. Both residential macrophages and T-cells are identified in the pancreas, and they are found to play critical roles in islet development and normal function [147,148,149]. We have also demonstrated that transient recruitment of T-helper 2 (Th2) lymphocytes and macrophages in IUGR fetal islets results in localized inflammation and leads to the development of T2D in adulthood [6]. Pancreatic islets are innervated, which is important for regulating islet hormone secretion and glucose metabolism [48]. Reduced islet innervation is associated with glucose intolerance and islet dysfunction in type 2 diabetes [49].

Interestingly, IUGR induced alteration of H3K4me3 and H3K27me3 modifications at many transcription factor binding motifs. These transcription factors were also identified as critical upstream regulators in our transcriptome study [9], and are known to play important roles in maintaining normal β-cell and islet function, such as the ETS family, Sox family, C/EBPs, Bcl6, Ebf1, Mitf, Nanog, Tcf, and Thrb. Several transcription factors without clearly understood function in islets, including Foxl2, Nkx2.5, and Pax7, were identified in both ChIP-seq and our transcriptome datasets. Foxl2 is required for ovary development and function, as well as somatic testis determination [150]. Foxl2 also regulates the expression of the steroidogenic acute regulatory (*StAR*) gene, which mediates the transport of cholesterol across the mitochondrial membrane and controls steroidogenesis [151]. Nkx2.5 is critical for cardiac development [152]. It also regulates the expression of fibroblast growth factor 16 (*Fgf16*), a growth factor for embryonic brown adipocytes [153,154]. Pax7 is important in neural crest development and gastrulation, and regulates the expression of neural crest markers, including Sox9 and Sox10 [155]. Pax7 also plays a role in skeletal muscle myogenesis [156]. The function of these transcription factors in islets is still unknown. The impact of their changes on islet function and IUGR phenotypes remains to be further investigated.

Interestingly, only six genes, *Trpm5, Tfam, Mcf2l, Slc28a2, Tnf*, and *Mpz* that had differential expression in IUGR versus control islets had changes in all three histone modifications. The low number of genes with altered expression that correlated with changes in the enrichment of all three marks was unexpected and suggested that perhaps there is not a histone code in differentiated β-cells in vivo.

Bivalent genes play critical roles in embryonic stem cells and during development [98,99]. Lu et al. have shown that loss of H3K27me3 marks at poised/bivalent domains may contribute to β-cell de-differentiation, dysfunction, and diabetes [104]. Our ChIP-seq data revealed more than 1000 potential bivalent genes in islets, and many of their poised states were altered by IUGR. These included genes important for β-cell function, such as *Grpr, Pcsk9, Ang2, Grm7*, and *Nr0b2*. Interestingly, we identified six potential bivalent genes, *Acod1, Fgf21*, *Serpina11, Cdh16, Lrrc27*, and *Lrrc66*, whose alterations in bivalency and expression levels may contribute to IUGR phenotypes observed in adult islets. Although the roles of *Serpina11, Cdh16, Lrrc27*, and *Lrrc66* in regulating islet function remain unclear, *Acod1* is known to be important in modulating the immune system and mitochondrial function [116,117,118], and *Fgf21* can protect against type 2 diabetes in mice by increasing insulin expression and secretion [119,122]. However, whether these genes are truly bivalent will need to be determined. Furthermore, how their dysregulation contributes to β-cell dysfunction, islet phenotype, and development of T2D in IUGR animals remains to be clarified.

Although animal studies cannot completely translate to human research, animal models have a normal genetic background upon which environmental effects during gestation or early postnatal life can be tested in vivo for their role in inducing diabetes. However, our current study using a rat model to investigate IUGR-induced histone modification changes and their association with β-cell dysfunction has certain limitations. First, rat genome is not as well-annotated as human or mouse genomes, thus the genomic and epigenetic changes observed in rats may be difficult to apply to the human genome. Second, due to tissue or species specificity, some IUGR-induced epigenetic changes in rat islets may not be the same as changes in other tissues or in humans. For example, in a human study of IUGR placenta, H3K27Ac modifications are altered in genes comprising HIF-1-alpha pathway and transcription factor binding motifs for SP1, ARNT2, HEY2, and VDR [136]. These changes were not found in our rat IUGR islets. Despite these limitations, the β-cell development in the rats is very similar to what has been observed in the human, which makes it a useful animal model to study the molecular mechanisms for IUGR-induced diabetes.

## 5. Conclusions

Histone modifications are an important epigenetic mechanism modulating gene expression changes and the development of T2D. This study is the first genome-wide study of histone modification of the marks H3K4me3, H3K27me3, and H3K27Ac in rat islets. While serving as a valuable resource for elucidating chromatin landscapes in rat islets, we have demonstrated alterations of histone modifications in key transcription factor binding motifs and genes as a potential epigenetic mechanism contributing to long-term gene dysregulation and abnormal islet phenotypes in IUGR rats.

## Figures and Tables

**Figure 1 cells-09-01435-f001:**
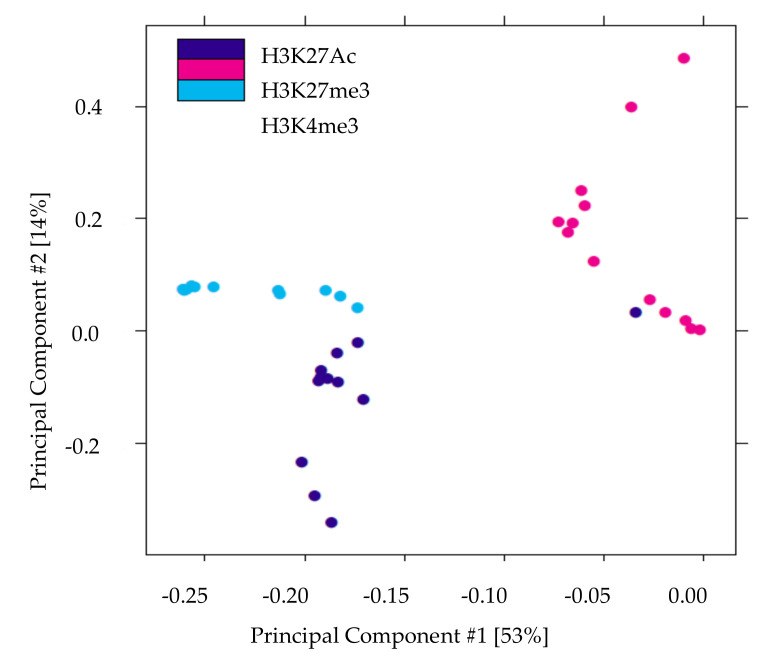
Principal components analysis of histone modifications in islets demonstrates that the three histone marks clearly separate. The X-axis (Comp1) represents 53% of the variability between samples and the Y-axis (Comp 2) represents 14% of the variability between samples.

**Figure 2 cells-09-01435-f002:**
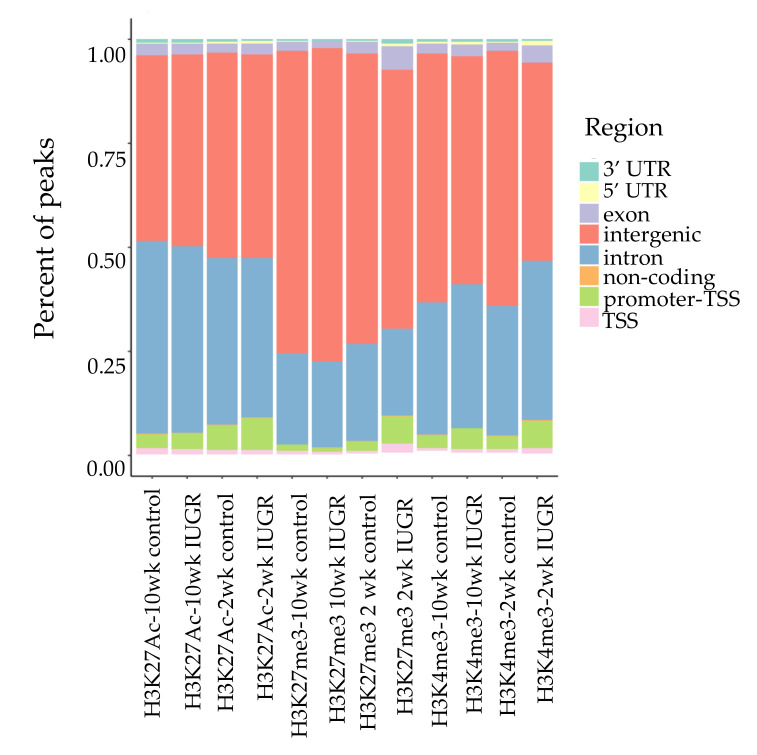
Genomic locations of histone modifications. Bars represent the location and proportion of each histone mark, TSS defined as 100 bp upstream to 1 kb downstream of TSS. Promoter-TSS is 1 kb upstream to 100 bp downstream of TSS. 5′ untranslated region (5 ‘UTR) starts at TTS and ends before the initiation codon. 3′ UTR starts immediately following termination codon. Non-coding regions are non-encoding protein sequences. Intergenic regions represent regions beyond the above defined regions.

**Figure 3 cells-09-01435-f003:**
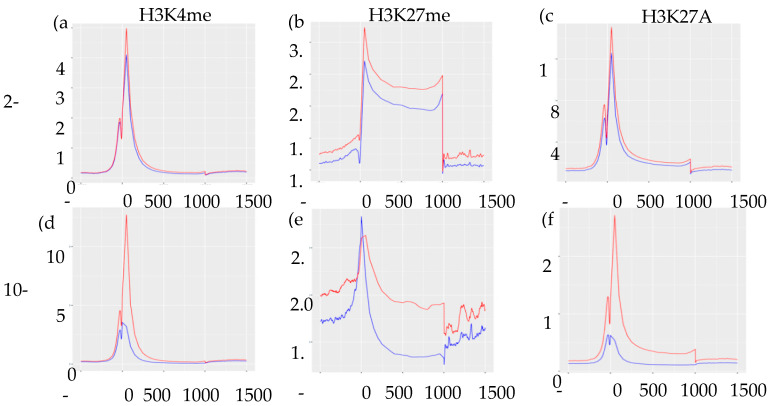
Transcription start site plots (TSS plots) of histone modifications showing the distribution and enrichment of each histone mark. (**a**–**c**) represents H3K4me3, H3K27me3, and H3K27Ac marks in 2-wk islets, respectively. (**d**–**f**) represents H3K4me3, H3K27me3, and H3K27Ac mark in 10-wk islets, respectively. Peak input/enrichment shown on vertical axes and position base-pair shown on horizontal axes. Red lines indicate control islets and blue lines indicate IUGR islets.

**Figure 4 cells-09-01435-f004:**
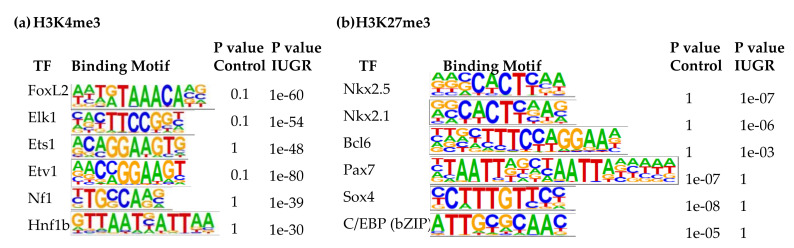
Transcription factor binding motifs enriched in islets at 2 weeks of age. (**a**) H3K4me3; (**b**) H3K27me3.

**Figure 5 cells-09-01435-f005:**
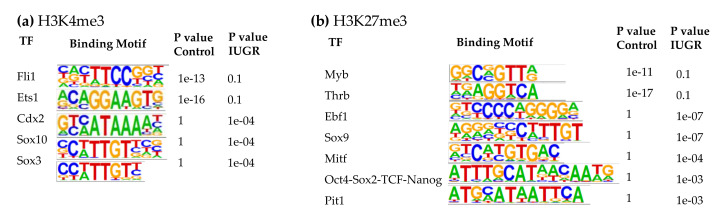
Transcription factor binding motifs enriched in islets at 10 weeks of age. (**a**) H3K4me3; (**b**) H3K27me3.

**Table 1 cells-09-01435-t001:** The number of genes potentially regulated by histone modifications.

	*2-wk*			*10-wk*		
Histone Mark	Total Gene #	Up-Regulated Gene #	Down-Regulated Gene #	Total Gene #	Up-Regulated Gene #	Down-Regulated Gene #
**H3K4me3**	41	29	12	638	344	294
**H3K27me3**	61	23	38	357	24	333
**H3K27Ac**	148	52	96	401	29	372
**All 3 marks**	0	0	0	6	3	3

**Table 2 cells-09-01435-t002:** Genes potentially regulated by all 3 marks.

Gene	Gene Name	H3K4me3 logFC	H3K27me3 logFC	H3K27Ac logFC	RNASeq logFC	RNASeq FDR
*Trpm5*	transient receptor potential cation channel, subfamily M, member 5	1.27	−0.78	0.96	0.8302	0.0215
*Tfam*	transcription factor A, mitochondrial	0.90	−0.95	1.10	0.5014	0.0148
*Mcf2l*	MCF.2 cell line derived transforming sequence-like	1.17	−1.06	1.19	0.4669	0.0500
*Slc28a2*	solute carrier family 28 member 2	−1.05	1.77	−0.75	−1.2177	0.0040
*Tnf*	tumor necrosis factor	−1.47	1.01	−0.95	−2.1019	0.0020
*Mpz*	myelin protein zero	−0.55	0.42	−0.56	−2.4936	0.0027

**Table 3 cells-09-01435-t003:** Differentially expressed genes with persistent changes for H3K27Ac histone mark.

	2-wk Islets			10-wk Islets		
Genes	RNAseq logFC	RNAseq FDR	H3K27Ac logFC	RNAseq logFC	RNAseq FDR	H3K27Ac logFC
*Bcat2*	−1.08	1.30 × 10^−2^	−0.98	−1.37	1.74 × 10^−4^	−0.88
*Bhlha15*	−2.12	8.77 × 10^−5^	−0.66	−1.42	5.75 × 10^−3^	−0.48
*Cyp2t1*	−0.99	1.14 × 10^−2^	−0.63	−0.78	3.54 × 10^−2^	−0.56
*Dram1*	−1.42	1.41 × 10^−3^	−0.12	−1.92	5.29 × 10^−7^	−0.76
*Foxa3*	−0.98	3.16 × 10^−2^	−0.67	−0.92	2.37 × 10^−2^	−0.47
*Gpt2*	−1.39	2.83 × 10^−4^	−0.59	−0.91	1.67 × 10^−2^	−0.77
*Mast3*	−1.10	2.69 × 10^−4^	−0.56	−0.83	4.53 × 10^−3^	−0.77
*Ppard*	−1.36	5.15 × 10^−5^	−0.06	−1.97	7.32 × 10^−11^	−0.74
*Pycr1*	−1.85	1.82 × 10^−4^	−0.87	−1.18	2.05 × 10^−2^	−0.82
*Reep5*	−0.92	4.49 × 10^−2^	−0.41	−1.48	2.62 × 10^−5^	−0.47
*Tex49*	−2.10	3.54 × 10^−3^	−0.33	−1.93	2.48 × 10^−2^	−1.35
*Rnh1*	−1.15	1.39 × 10^−2^	−0.28	−1.97	1.32 × 10^−7^	−0.56
*Rnpepl1*	−1.03	1.17 × 10^−3^	−0.64	−1.37	9.37 × 10^−7^	−0.48
*Sel1l*	−1.61	3.84 × 10^−3^	−0.38	−1.86	9.29 × 10^−5^	−0.95
*Sfxn2*	−1.42	2.34 × 10^−3^	−0.51	−0.97	3.91 × 10^−2^	−0.90
*Tnip2*	−0.61	3.89 × 10^−2^	−0.55	−0.59	2.96 × 10^−2^	−0.59
*Tst*	−1.26	1.41 × 10^−2^	−0.14	−3.13	9.28 × 10^−14^	−0.54
*Ulk1*	−1.03	2.87 × 10^−5^	−0.41	−0.70	4.26 × 10^−3^	−0.66
*Wdtc1*	−0.92	2.98 × 10^−3^	−1.01	−0.90	1.26 × 10^−3^	−0.51
*Xbp1*	−1.14	1.64 × 10^−2^	−0.56	−1.16	4.90 × 10^−3^	−0.84

**Table 4 cells-09-01435-t004:** Top IPA pathways potentially regulated by histone modifications and disrupted in 10-wk IUGR islets.

Histone Mark	Ingenuity Canonical Pathways	*p*-Value
H3K4me3	Type II Diabetes Mellitus signaling	1.32 × 10^−4^
	antiproliferative role of TOB in T- cell signaling	7.24 × 10^−4^
	Tight junction signaling	1.00 × 10^−3^
	RAR activation	1.12 × 10^−3^
	PI3K signaling in B lymphocytes	1.23 × 10^−3^
	Factors promoting cardiogenesis in vertebrates	1.23 × 10^−3^
	BMP signaling pathway	1.41 × 10^−3^
	Neurotrophin/TRK signaling	1.51 × 10^−3^
	PPAR signaling	1.55 × 10^−3^
H3K27me3	Cellular stress and injury	2.24 × 10^−7^
	Hepatic fibrosis / hepatic stellate cell activation	1.41 × 10^−6^
	cAMP-mediated signaling	1.32 × 10^−5^
	Macrophage function	2.95 × 10^−5^
	Mitochondrial function / Autophagy	3.55 × 10^−5^
	Citrulline-nitric oxide cycle	3.63 × 10^−5^
	Wnt/β-catenin signaling	8.13 × 10^−5^
	Axonal guidance signaling	9.12 × 10^−5^
	eNOS signaling	3.63 × 10^−4^
H3K27Ac	Polyamine regulation	1.78 × 10^−6^
	Cell migration signaling	1.02 × 10^−5^
	Production of nitric oxide and reactive oxygen species in Macrophages	1.05 × 10^−5^
	IL-9 signaling	1.35 × 10^−5^
	PDGF signaling	3.24 × 10^−5^
	PPAR signaling	4.90 × 10^−5^
	Unfolded protein response	5.13 × 10^−5^
	ERK/MAPK signaling	6.46 × 10^−5^
	Phospholipase C signaling	1.12 × 10^−4^

**Table 5 cells-09-01435-t005:** Transcription factor binding motifs with persistent histone modification changes.

H3K4me3					
Motif Name	Consensus	2w_IUGR *p*-Value	2w_Ctrl *p*-Value	10w_IUGR *p*-Value	10w_Ctrl *p*-Value
ELF3	ANCAGGAAGT	1.00 × 10^−43^	0.1	1.00 × 10^−9^	1
ELF5	ACVAGGAAGT	1.00 × 10^−34^	0.1	1.00 × 10^−7^	1
ERG	ACAGGAAGTG	1.00 × 10^−71^	0.1	1.00 × 10^−8^	1
ETS1	ACAGGAAGTG	1.00 × 10^−48^	1	1.00 × 10^−16^	0.1
ETV1	AACCGGAAGT	1.00 × 10^−80^	0.1	1.00 × 10^−11^	1
ETV2	NNAYTTCCTGHN	1.00 × 10^−62^	1	1.00 × 10^−9^	0.1
EWS:ERG	ATTTCCTGTN	1.00 × 10^−69^	0.1	1.00 × 10^−8^	1
GABPA	RACCGGAAGT	1.00 × 10^−58^	0.1	1.00 × 10^−8^	0.1
SMAD4	VBSYGTCTGG	1.00 × 10^−24^	1	1.00 × 10^−6^	1
**H3K27me3**					
**Motif name**	**Consensus**	**2w_IUGR** ***p*-Value**	**2w_Ctrl** ***p*-Value**	**10w_IUGR** ***p*-Value**	**10w_Ctrl** ***p*-Value**
MEIS1	VGCTGWCAVB	1	1.00 × 10^−4^	1	1.00 × 10^−17^
PR	VAGRACAKNCTGTBC	0.1	1.00 × 10^−11^	1	1.00 × 10^−20^
OLIG2	RCCATMTGTT	1	1.00 × 10^−5^	0.1	1.00 × 10^−10^

**Table 6 cells-09-01435-t006:** Critical upstream regulators identified in transcriptome datasets.

2-wk			
Upstream Regulator	Activation z-Score	*p*-Value	Target Gene Number
FOXL2	2.06	7.02 × 10^−3^	7
ELK1	0.74	1.77 × 10^−3^	7
ETS1	0.36	1.65 × 10^−2^	14
NKX2-5	−0.39	4.13 × 10^−2^	4
BCL6	−1.24	5.37 × 10^−3^	12
CEBPA	1.89	1.87 × 10^−7^	37
CEBPB	2.73	1.94 × 10^−7^	36
PAX7	1.14	8.16 × 10^−4^	9
SOX4	1.89	5.68 × 10^−4^	16
HNF1B	−1.63	5.26 × 10^−5^	14
**10-wk**			
**Upstream Regulator**	**Activation** **z-Score**	***p*-Value**	**Target Gene Number**
FLI1	0.45	5.42 × 10^−4^	14
CDX2	−0.84	1.29 × 10^−4^	26
SOX10	−0.84	1.80 × 10^−6^	13
EBF1	−1.23	1.09 × 10^−5^	30
SOX9	−0.43	3.72 × 10^−2^	11
MITF	−3.25	8.96 × 10^−4^	39
NANOG	−0.24	2.62 × 10^−3^	22
TCF	−2.00	9.48 × 10^−8^	27
THRB	0.42	2.38 × 10^−8^	47
ELF3	−2.07	5.52 × 10^−3^	8
ERG	−4.45	1.40 × 10^−10^	50

**Table 7 cells-09-01435-t007:** Potential bivalent genes critical for long-term IUGR phenotype.

Gene	Gene Name	Bivalency			
		2-wk		10-wk	
		Control	IUGR	Control	IUGR
*Acod1*	aconitate decarboxylase 1	+	+	−	+
*Fgf21*	fibroblast growth factor 21	+	+	−	+
*Serpina11*	serpin family A member 11	+	+	−	+
*Cdh16*	cadherin 16	−	−	+	−
*Lrrc27*	leucine-rich repeat-containing 27	−	−	+	−
*Lrrc66*	leucine-rich repeat-containing 66	−	−	+	−

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
