# Peer review of "Altered Transcription Factor Binding and Gene Bivalency in Islets of Intrauterine Growth Retarded Rats"

_cells, 2020, doi:10.3390/cells9061435_

Round 1
Reviewer 1 Report
The optimal perfusion of a placenta is critical for the development of a fetus. Intrauterine growth retardation (IUGR) of the fetus is one consequence of an under perfused placenta. IUGR is associate with long term health consequences for the baby such as diabetes and cardiovascular disease. The manuscript by Lien at al examines the epigenetic modifications and changes in gene expression in islets from 2 and 10 week old rats. ChIP-Seq analysis showed that IUGR-induced histone mark changes were enriched at critical transcriptional binding motifs and potential bivalent genes were altered in IUGR islets.
The manuscript is well organized and easy to follow. I enjoyed reading it although at times felt a bit overwhelmed with all the tables. I would have enjoyed seeing some H3K4me3, H3K27me3 and H3K27Ac tracks between the 2wks vs 10wks data of some of the critical loci in the manuscript itself. Overall, it is a great resource for the research community.
Some minor points.
- Do the 6 potential bivalent genes identified in table 7 show changes in all three histone modifications or just the H3K4me3 and H3K27me3?
- How many islet cells were used for ChIP-Seq?
- Supplemental tables 6- 9 are missing. I was only able to view Supplemental tables 1-5.
4. Can table 3 be presented as a heatmap?
Author Response
We thank the Reviewers for their comments and suggestions. Our revised manuscript is improved in clarity and substance as a result of their comments and suggestions.
Reviewer 1:
- Do the 6 potential bivalent genes identified in table 7 show changes in all three histone modifications or just the H3K4me3 and H3K27me3?
Response: Only Fgf21 and Lrrc27 showed H3K27Ac changes in 10-wk islets. For Fgf21, H3K27Ac modification decreased (logFC = -0.56) in 10-wk IUGR islets near the bivalent domain. For Lrcc27, there was a decreased H3K27Ac mark (logFC = -0.68) in the loci 4.5 Kb downstream of the bivalent domain in 10-wk IUGR islets.
- How many islet cells were used for ChIP-Seq?
Response: We did not count the islet number. All the islets isolated from 2-4 animals were pooled to prepare chromatin, and 4-5 µg of chromatin were used for each ChIP with antibody for histone modification.
- Supplemental tables 6- 9 are missing. I was only able to view Supplemental tables 1-5.
Response: We are sorry that reviewers were not able to view Supplemental tables S6-S9. We submitted all supplemental tables together in an excel file containing 9 spread sheets, and didn’t know why supplemental tables S6-S9 were missing after they were automatically converted to the PDF file by MDPI system. In the revised manuscript, we separated and submitted supplemental tables in two excel files, Supplemental Tables S1-S5 and Supplemental Tables S6-S9, to avoid this happening again.
- Can table 3 be presented as a heatmap?
Response: We decided to present the data in a table because we think it is more easily understood and provides more details than a heatmap.
Reviewer 2 Report
My congratulations to authors for well written manuscript.
Author Response
- the limitations of the study should be discussed in detail in an aspect of the difference between a rat model and human cases. For example, there is a paper investigated the changes in H3K27Ac and gene expression and their relationship with fetal growth restriction using human placenta biopsies. (https://doi.org/10.1186/s13148-018-0508-x).
Response: The H3K27 acetylation study in human placenta from fetal growth restriction by Paauw et al. has been cited as reference #136 in Ln437 in the revised manuscript. The discussion of limitations for rat model in studying intrauterine growth restriction and diabetes has been added in the discussion section from Ln513 - Ln525.
Reviewer 3 Report
The authors investigated the epigenetic modifications and their potential effects on permanent changes in gene expression using Intrauterine growth retardation (IUGR) rat model by ChIP-seq analysis. The enrichment of histone mark changes at significant transcription factor binding motifs was found and target gene candidates to long-term gene dysregulation and islet abnormality in islets. The findings are helpful to understand the effect of IUGR in an aspect of the development of T2D. However, the limitations of the study should be discussed in detail in an aspect of the difference between a rat model and human cases. For example, there is a paper investigated the changes in H3K27Ac and gene expression and their relationship with fetal growth restriction using human placenta biopsies. (https://doi.org/10.1186/s13148-018-0508-x)
Author Response
Editor:
- I suggest the authors could also cite (maybe in lines 436-438) the work demonstrating the generational effects of such epigenetic modifications in rats, as demonstrated in the PMID #26166746 (Cell Metab. 2015).
Response: The multigenerational study by Hardikar et al. (PMID #26166746) has been cited as reference #135 in Ln437 and Ln438 in the revised manuscript.